# Sex-Specific Effects of Dietary Methionine Restriction on the Intestinal Microbiome

**DOI:** 10.3390/nu12030781

**Published:** 2020-03-16

**Authors:** Katherine F. Wallis, Stepan B. Melnyk, Isabelle R. Miousse

**Affiliations:** 1Department of Biochemistry and Molecular Biology, University of Arkansas for Medical Sciences, Little Rock, AR 72205, USA; KFWallis@uams.edu; 2Arkansas Children’s Research Institute, Little Rock, AR 72202, USA; MelnykSB@archildrens.org

**Keywords:** methionine, microbiome, methylation, diet

## Abstract

Dietary methionine restriction is associated with improved health outcomes and an increase in lifespan in animal models. We have previously shown that an increase in dietary methionine induces alteration in the intestinal microbiome. The composition of the intestinal microbiota is a determinant of health and we, therefore, hypothesized that dietary methionine restriction would also induce changes in the murine microbiome. After one month on a methionine-restricted diet, five-month-old male and female C57BL/6 mice had decreased levels of serum methionine, without changes in body weight. We identified a decrease in the hepatic methylation status of animals fed a methionine-restricted diet compared to controls. This decrease was not associated with changes in DNA or protein methylation in the liver. In males, we saw an increase in families *Bacteroidaceae* and *Verrucoccaceae* (mostly *A. mucinophila*) and a decrease in *Rumminococcaceae* in animals fed a methionine-restricted diet compared to controls. In females, *Bacteroidales* family S24-7 was increased two-fold, while families *Bacteroidaceae*, *Verrucoccaceae, Rumminococcaceae,* and *Rikenellaceae* were decreased compared to controls. In summary, feeding a methionine-restricted diet for one month was associated with significant and sex-specific changes in the intestinal microbiome.

## 1. Introduction

Methionine is an essential amino acid used not only for protein synthesis but also as a direct precursor for S-adenosylmethionine (SAM), the universal methyl donor for DNA and proteins, including histones. Animal sources such as meat and dairy generally provide all nine essential amino acids (methionine, histidine, isoleucine, leucine, lysine, phenylalanine, threonine, tryptophan, and valine), referred to as complete proteins. Most plant proteins are too low in methionine to be considered complete. Increasingly, individuals elect to follow diets that are rich in animal, complete proteins. The appeal of low carbohydrate, high protein diets is attributable to the role of protein in muscle anabolism [1,2], the negative perception of both fatty and sugary food in the general public [3], and to the satiating effect of dietary protein, regarded as an aid in weight loss [4,5]. In parallel, plant-based diets are also gaining popularity, for environmental [6], social [7], or health-related reasons. Plant-based diets have been associated with a reduced cancer incidence [8] and may be helpful in the prevention and treatment of type II diabetes [9].

Complete methionine deprivation leads to rapid weight loss, hepatosteatosis, and leukopenia [10,11]. However, methionine restriction to a level of 0.12–0.15% in rodent diets (compared to 0.60–0.86% in a standard diet, and similar to a plant-based diet [12]) is consistently associated with an increase in lifespan in animal models [13,14,15,16,17]. It is also associated with many health benefits, such as improvements in insulin and lipid regulation and barrier function [18]. Furthermore, methionine restriction significantly impairs tumor growth and metastasis in preclinical models [10,19]. In yeast, lifespan extension by methionine restriction is dependent on mitophagy, a subtype of autophagy where damaged mitochondria are targeted for recycling in the lysosome [15]. In mammals, dietary methionine levels may act through additional mechanisms, such as modifications of intestinal microbial populations.

Advances in sequencing technologies over the past 20 years have led to the development of research on the interaction between hosts and their microbial flora. New connections have been drawn between the composition of the microbial populations in the gut and human health (reviewed in [20]). Fecal transplant experiments in germ-free mice have established causal links between the composition of the gut microbiome and health outcomes, as exampled by diet-induced obesity [21] or the responsiveness to immune checkpoint inhibitors [22,23,24]. This brings the question of how to induce a favorable microbiome. Consumption of dietary fibers, for example, is strongly associated with health benefits (reviewed in [25]). Methionine supplementation interacts with the prokaryotic populations of the gut [26]. In rodents, high levels of dietary methionine mimicking those seen in the Western diet (1.95% per weight) were found to alter the intestinal microbiome [26]. These changes were associated with damage to the gut epithelium, a loss of expression of tight junction proteins, and translocation of bacterial genetic material to the liver [26].

The impact of methionine restriction on the gut microbiome has not been reported yet. Therefore, we hypothesized that dietary methionine restriction would also affect the gut microbiome. We present here our results describing the effect of dietary methionine restriction for a period of 30 days in male and female adult mice on the gut microbiome.

## 2. Materials and Methods

### 2.1. Animal Studies

This project was approved by the Institutional Animal Care and Use Committee at the University of Arkansas for Medical Sciences (IACUC #3874). C57BL/6 mice, originally from Jackson Laboratories (Bar Harbor, ME, USA), were bred on-site at the University of Arkansas for Medical Sciences. Mice were kept under standardized conditions with controlled temperature and humidity and a 12 h light/dark cycle. Mice were 5 months of age at the beginning of the experiment and were fed an identical standard laboratory chow up to the start of the experiment. Each group contained 5 animals, except for the males on the methionine-restricted diet with 4 animals. Diets were changed at the beginning of the experiment to a methionine-restricted diet containing 0.15% methionine, or an otherwise identical control diet containing 0.65% per weight methionine (Teklad Diets TD.160240 andTD.140520, Appendix A). Diets were maintained continuously for 30 days. At the end of the study period, mice were anesthetized with isoflurane and retroorbital bleeding was performed. The collected blood samples were allowed to coagulate at room temperature then centrifuged at 10,000× *g* for 10 min at 4 °C. The serum was transferred to new tubes and flash-frozen until analysis. The right lobe of the liver and cecum content were excised and flash-frozen.

### 2.2. Metabolites

Serum and liver concentrations of metabolites in the methionine cycle were assessed as previously described [26]. Briefly, metabolites in the one-carbon pathway were measured at the Core Metabolomics Laboratory located at the Arkansas Children’s Research Institute by High Performance Liquid Chromatography (HPLC) coupled with coulometric detection and normalized to serum volume or liver tissue weight.

### 2.3. LINE-1 DNA Methylation

DNA was extracted from ~20 mg of flash-frozen liver tissue according to the manufacturer’s instructions (AllPrep, Qiagen, Germantown, MD, USA). Four biological replicates were used in each diet group for each sex. A total of 500 ng of g DNA was digested with 0.5 U each of 5 methylation-sensitive enzymes (SmaI, HhaI, HpaII, AciI, and BstUI (New England Biolabs, Ipswich, MA, USA)), as described previously [27]. In parallel, 500 ng of g DNA was sonicated for 5 min on a high setting as a control (Bioruptor, Diagenode, Denville, NJ, USA). The resulting DNA was diluted to 2.5 ng/µL. The 5’UTR of the repetitive element LINE-1 was PCR-amplified from a total of 10 ng of digested and sonicated DNA, with four technical replicates, using the following primers: forward AGTGGATCACAGTGCCTGC, reverse GGGTAGCCTGCTTCCCTATG. The abundance of methylated DNA (not cut by the methylation-sensitive enzymes) was normalized to the abundance of sonicated DNA in that same region and fold change calculated following the ΔΔCt method.

### 2.4. Immunodetection of Methylated Proteins

Samples were prepared from ~20mg of flash-frozen liver tissue in radioimmunoprecipitation assay (RIPA) buffer. The lysates were mixed with Laemmli sample buffer, boiled for 10 min, and 20 µg of each protein lysate was loaded onto the lanes of a 4–12% bis-tris gel (NuPage, ThermoFisher, Waltham, ME, USA). Proteins were then transferred onto a low fluorescence polyvinylidene difluoride (PVDF) membrane (Bio-Rad, Hercules, CA, USA) and blocked with 1% bovine serum albumin (BSA) in tris-buffered saline with 0.1% Tween-20 (TBST). The membrane was probed with a rabbit polyclonal antibody against methylated lysines (ab 23366, Abcam, Cambridge, ME, USA). Detection was performed with a DyLight 800 sheep anti-rabbit IgG secondary antibody on a ChemiDoc instrument (Bio-Rad, Hercules, CA, USA). An identical membrane was stained with amido black as a control.

### 2.5. Microbiome Analysis

Microbiome genomic analysis of the gut of mice was performed by use of the LoopSeq™16S Microbiome SSC 24-Plex kit (Loop Genomics, San Jose, CA, USA). The Loop protocol uses unique molecular labeling of individual 16S genes. This unique molecular identifier is then distributed throughout the gene, allowing fragmentation and sequencing by short reads on an Illumina platform, with subsequent reassembly of full-length 16S genes. Since the entire 16S gene is sequenced, all 9 (V1–V9) variable regions are identified, allowing the identification and relative abundance of bacteria at the species level.

Sequencing libraries were constructed from 10 ng of genomic DNA extracted from mouse gut microbiome according to the protocol supplied by the manufacturer. Libraries were sequenced on an Illumina NextSeq 500 platform (Illumina, San Diego, CA, USA), using a mid-output flow cell and paired-end 2 × 150 bp reads. Coverage was 100-150M paired-end reads per library of 24 samples. Raw sequencing data were collected in real time on Illumina’s BaseSpace, which generates FastQ files, which were then uploaded to Loop’s analytic pipeline.

Analytic results included CSV files containing single molecule abundance based on unique molecular identifiers (UMI’s), reference based (Silva) taxonomic classification based on the nine 16S variable regions and long reads of the whole 16S gene, OTU tables, rarefaction analysis, and RAD analysis.

### 2.6. Statistical Analysis

For metabolites and sequencing, groups were compared using a two-way ANOVA followed by Sidak’s multiple comparisons test to compare the two diets within each sex. Analyses were performed using the GraphPad Prism software version 8.3.0 (GraphPad software, San Diego, CA, USA).

## 3. Results

### 3.1. Dietary Methionine Restriction Alters Methionine-Related Metabolites

After 30 days on a diet containing 0.15% per weight methionine (23% of the level found in a control, methionine adequate diet (MAD)), we observed no difference in body weight between the two diets in either males or females (Figure 1A). We confirmed the validity of the diet by measuring methionine and its metabolites in the serum and liver of treated animals. We saw significant decreases in serum methionine concentration in both male and female groups (Figure 1B) with no significant difference in either free reduced homocysteine or homocystine (Figure 1C and Appendix A). The serum concentration of S-adenosylhomocysteine (SAH) was equally unchanged (Appendix A). However, the ratio of S-adenosylmethionine (SAM) to SAH (“methylation ratio”) was significantly decreased in females, but not in males (Figure 1D) and was driven by a significantly decreased level of SAM. The ratio of free reduced glutathione (GSH) to oxidized glutathione (GSSG) (“oxidative stress ratio”) was decreased in males, indicating an increase in oxidative stress (Figure 1E).

In liver, a tissue where the methionine cycle is tightly regulated, there was a significant decrease in methionine levels in females, but the change did not reach significance in males (Figure 1F). Liver free homocysteine level was decreased in both male and female groups (Figure 1G). SAM level in liver was decreased in males only, with no difference in SAH in either sex (Appendix A). The SAM/SAH ratio in liver was significantly decreased in both male and female groups (Figure 1H), suggesting a decreased global methylation capacity in the animals fed a methionine-restricted diet. The GSH/GSSG ratio in liver was decreased, reaching significance only in females (Figure 1I).

### 3.2. Hepatic DNA and Protein Methylation Is Unchanged by Methionine Restriction

Methionine is the precursor for SAM, the methyl donor for both DNA and protein methylation. We assessed the impact of dietary methionine restriction on the methylation of DNA and cellular proteins. We measured DNA methylation in the 5’UTR of the LINE-1 repetitive element. LINE-1 repeats cover ~20% of the mammalian genome and the 5’UTR is heavily methylated to preserve DNA stability. We did not detect any difference in DNA methylation in the 5’UTR of the LINE-1 element associated with the decrease in hepatic methionine concentration induced by the diet (Figure 2A). Similarly, we did not detect any change in the abundance of mono- and di-methylated lysine residues in total lysate from hepatic tissues (Figure 2B–D).

### 3.3. Dietary Methionine Restriction Has Sex-Specific Effects on the Microbiome

We performed a shotgun metagenomics analysis of the full-length 16S rRNA isolated from the cecum of animals on the methionine adequate or methionine-restricted diets. When sexes were combined, no differences were identified in the composition of the intestinal microflora between the two diets. Sex differences in microbiome composition have been well established both in human and in mice [28,29]. We, thus, reanalyzed the data segregating by sex (Figure 3A). Out of the six most abundant microbial families in the cecal samples analyzed, one showed no significant difference (*Lachnospiraceae*, Figure 3B), and five were significantly altered by the diet (*Bacteroidaceae*, bacteria of the *Bacteroidales* order but of unclassified family, *Ruminococcaceae*, *Rikenellaceae*, and *Verrucomicrobiaceae*).

One unclassified family of the *Bacteroidales* order represented around 20% of the total population in both males and females. The species “*Bacteroidales* S24-7 group uncultured bacterium” accounted for nearly 99% of that observation (Figure 3C). It was significantly increased, by nearly two-fold, in females on the methionine-restricted diet compared to control females and was unchanged in males. Similarly, abundance of the family *Rikenellaceae* did not change with diet in males, but it represented a significantly smaller percentage of the total population in females on a methionine-restricted diet than on a control diet (Figure 3D).

Bacteria of the *Bacteroidaceae* family, including the commensal obligate anaerobic Gram-negative bacteria *Bacteroides thetaiotaomicron* and *Bacteroides fragilis*, also differed in abundance between the two diets (Figure 3E). In females, the methionine-restricted diet was associated with a smaller total percentage of the population represented by *Bacteroidaceae*, while it was associated with a greater percentage in males.

For animals on a methionine-restricted diet, *Rumminococcaceae* represented a greater percentage of the total population in females and a smaller percentage of the population in males (Figure 3F). This family included genera such as *Ruminiclostridium* (Figure 3G), *Anaerotruncus* (Figure 3H), and *Oscillibacter* (Figure 3I). Only the genus *Ruminiclostridium* was significantly different between the two diets, in addition to showing an interaction between sex and diet. The genus *Anaerotruncus* was associated with sex only. The genus *Oscillibacter* was also associated with sex, and there was a significant interaction between sex and diet.

Finally, abundance of the family *Verrucomicrobiaceae* also showed a strong interaction between sex and diet (Figure 3J). This family contains the genus *Akkermansia* (Figure 3K), and particularly the species *Akkermansia mucinophila*. In humans, *A. mucinophila* is more abundant in females than in males and is associated with positive health outcomes [30]. In agreement with these results, *Akkermansia* represented a larger percentage of the total control population in female than in male mice in our experiment. The family *Verrucomicrobiaceae*, and *Akkermansia* in particular, was decreased by the methionine-restricted diet in females and increased in males.

## 4. Discussion

In this study, we looked at the effect of dietary methionine restriction on one-carbon metabolism and the intestinal microbiome in male and female C57BL/6 mice after one month. We did not measure any significant change in body weight. As expected, the methionine-restricted diet containing 0.15% methionine led to a significant decrease concentration of methionine in serum in both sexes. Despite lowering methionine concentration in serum, a methionine-restricted diet did not significantly affect serum concentration of methionine cycle metabolites, such as SAH, or any form of homocysteine. However, the decrease in serum SAM induced a significant decrease in the SAM/SAH ratio, in females only. A decrease in this ratio, also referred to as the methylation index or methylation status, is associated with atherosclerosis in mice [31]. Cardiac function is not affected by long-term methionine restriction in male mice [32], but studies in females are currently lacking. Significantly lower levels of homocysteine were found in liver in both groups after methionine restriction. Homocysteine is used for both the synthesis of methionine through the remethylation pathway and the synthesis of cysteine and GSH through the transsulfuration pathway. We also measured a significant decrease in the SAM/SAH methylation ratio in the liver in both sexes.

To evaluate the impact of this decrease in the hepatic SAM/SAH ratio, we measured DNA and protein methylation. Methylation in the LINE-1 5’UTR plays a key role in preventing retrotransposition events that can lead to chromosomal rearrangements, and eventually to cancer [33]. We did not identify any significant change in hepatic DNA methylation in that region after 30 days on a methionine-restricted diet. This is in good agreement with a previous report indicating that there was no change in hepatic LINE-1 methylation 12 weeks after the initiation of a methionine-restricted diet [34]. Methionine restriction is reported to reduce specifically H3K4me3, with little effect on other histone methylation marks [35,36]. In agreement with these results, we did not identify any change in the abundance of mono- and di-methylated lysines in total cellular proteins. The decrease in the hepatic SAM/SAH ratio, therefore, did not correlate with changes in these endpoints. Although this does not preclude possible changes in different endpoints or organs, it supports the idea that homeostasis is maintained under dietary methionine restriction.

Finally, we addressed the effect of a methionine-restricted diet on the intestinal microbiome. Sex differences in the microbiome have been well described [29], underlying our choice to analyze both males and females. Our analysis revealed large differences in microbiome composition between males and females on the standard diet. We observed that for at least three families of bacteria (*Bacteroidaceae*, *Rumminococcaceae*, and *Verrucomicrobiaceae*), the direction of the change was opposite in males versus females. The presence of *A. mucinophila* is associated with positive health outcomes in many reports (reviewed in [37]). The increase in abundance of *A. mucinophila* in males is, therefore, in agreement with the lifespan extension properties attributed to dietary methionine restriction in male [16] and female mice [17]. The decrease in *A. mucinophila* in females was, therefore, an unexpected finding. Also in females, we observed an increase in the species “Bacteroidales S24-7 group uncultured bacterium”. This species has been well described in the murine gut, although its significance to long term health is still unclear [38]. In males, we saw an increase in the abundance of the family *Bacteroidaceae* (phylum Bacteroidetes) and a decrease in the family *Ruminococcaceae* (phylum Firmicutes) associated with the methionine-restricted diet. This pattern of increased Bacteroidetes to Firmicutes ratio is associated with leanness in humans [39,40]. We observed the opposite in females, where there was a decrease in *Bacteroidaceae* and an increase in *Rumminococcaceae* compared with the control diet.

Interestingly, the Bacteroidetes to Firmicutes ratio is also correlated with alterations in gene-specific DNA methylation in blood [41]. This suggests that changes in methionine intake also have indirect effects on DNA and protein methylation mediated through the synthesis of compounds by the gut microbiome, such as folate. It also highlights the interconnections between diet, gut microbiome, epigenetic regulation, and health.

We showed that exposure to a methionine-restricted diet altered the murine intestinal microbiome in both males and females, in a sex-specific manner. The specific families and species of organisms identified here are representative of the animal housing environment where our experiment was conducted [42,43]. Experiments conducted at different sites may provide additional information about the microbial trends associated with a methionine-restricted diet and provide a clearer understanding of the health consequences associated with this diet.

## Figures and Tables

**Figure 1 nutrients-12-00781-f001:**
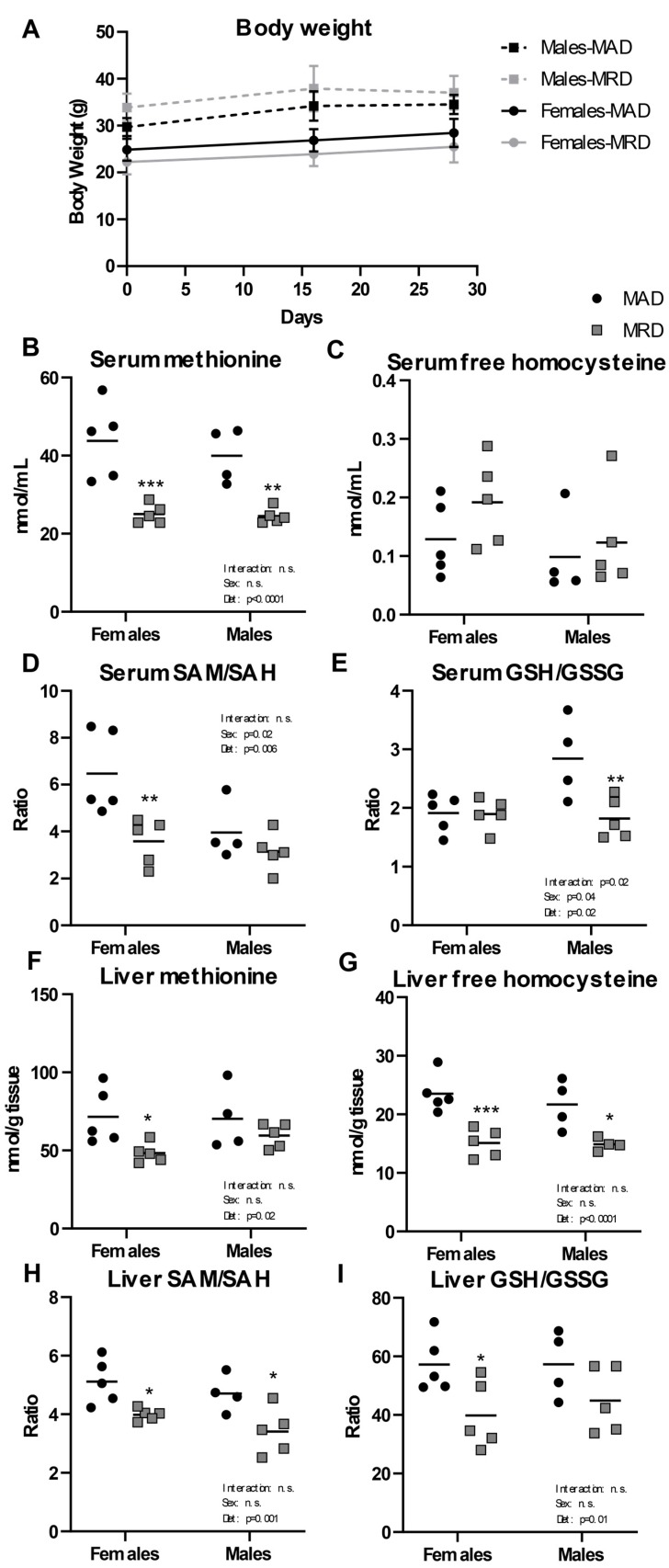
Body weight and metabolites. Panel (**A**). Male and female mice were weighed the day before initiation of the diet (Day 0), as well as at day 16 and 30. Panels (**B**–**I**). Metabolites were measured in serum and liver. Mean ± SEM, MAD: methionine adequate diet, MRD: methionine-restricted diet. Data were analyzed using a two-way ANOVA followed by Sidak’s multiple comparisons test to compare the two diets within each sex. * indicates *p* ≤ 0.05, ** indicates *p* ≤ 0.01, and *** indicates *p* ≤ 0.001. n.s. = Non significant.

**Figure 2 nutrients-12-00781-f002:**
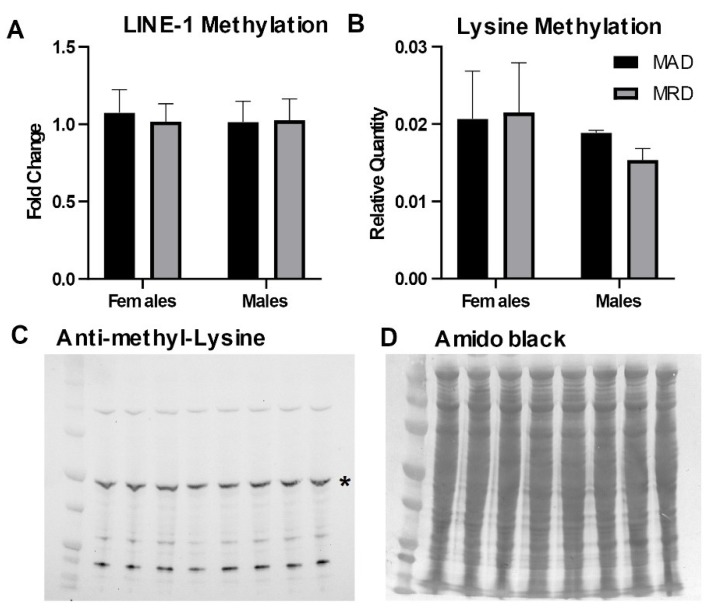
DNA and protein methylation. Panel (**A**). DNA methylation in the 5’UTR of the LINE-1 repetitive element measured by methylation-sensitive digestion followed by PCR amplification. Panels (**B**–**D**): Immunoblot against mono- and di-methylated lysines. The asterisk (*) in panel (**C**) indicates the band analyzed in panel (**B**). Mean ± SEM, MAD: methionine adequate diet, MRD: methionine-restricted diet. Data were analyzed using a two-way ANOVA followed by Sidak’s multiple comparisons test to compare the two diets within each sex.

**Figure 3 nutrients-12-00781-f003:**
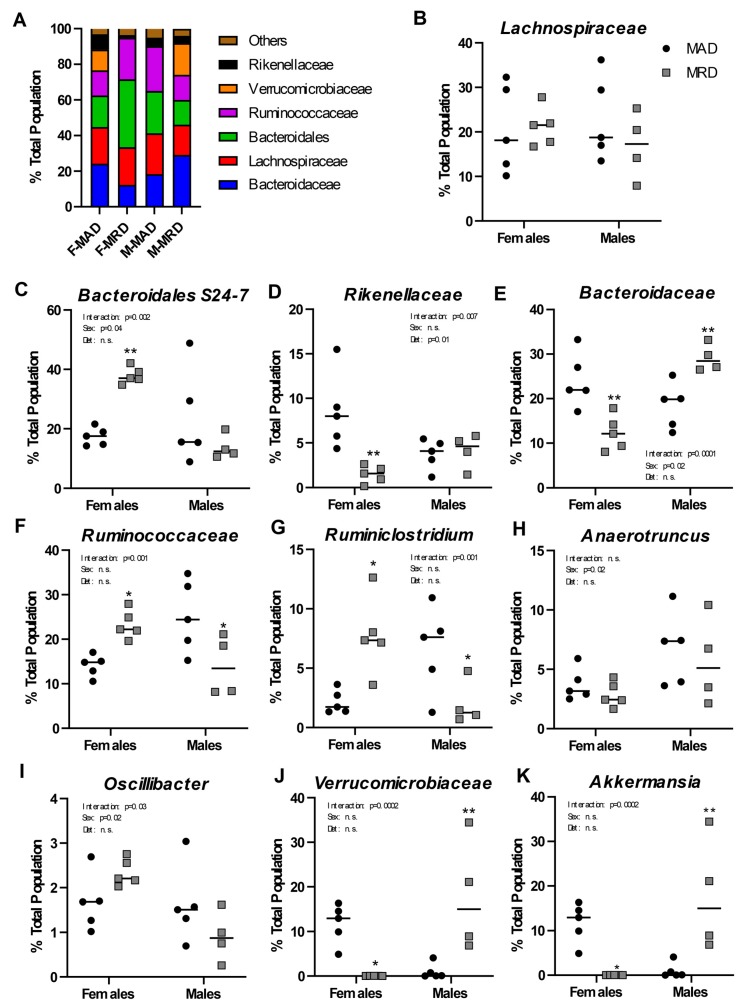
Changes in microbial populations. Panel (**A**). Overview of the bacterial families present in samples from each group as a function of % of the total population. Panels (**B**–**K**). Abundance of bacterial families (**B**,**D**–**F**,**J**) or genera (**C**,**G**–**I**,**K**) in male and females by diet group. Mean ±SEM, MAD: methionine adequate diet, MRD: methionine-restricted diet. Data were analyzed using a two-way ANOVA followed by Sidak’s multiple comparisons test to compare the two diets within each sex. * indicates *p* ≤ 0.05, ** indicates *p* ≤ 0.01, and *** indicates *p* ≤ 0.001. n.s. = Non significant.

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
