# Peer review of "Sex-Specific Effects of Dietary Methionine Restriction on the Intestinal Microbiome"

_nutrients, 2020, doi:10.3390/nu12030781_

Round 1

Reviewer 1 Report

The study by Wallis and colleagues (# nutrients-736856) highlights the prominent role of dietary methionine on the composition of the intestinal murine microbiome. The authors confirm previous findings and show a significant modulation of bacterial species in case of methionine restriction. Most prominent, these changes were dictated by the sex of the mice, while the protein methylation rate was stable. These finding speak for a well-controlled intracellular homeostasis in methionine metabolism.

The manuscript is clearly structured and reports scientifically new knowledge. However, the following points should be reconsidered and revised.

Comments:

  1. The number of animals per group is small (n = 5 or 4), which might lead to an overrepresentation of high or low metabolite values of individuals. Therefore, please present the data in figure 1B – 1I and in figure 3B – 3K as box plots, which allow the illustration of all individual values in each group.
  2. The heading of figure 1C should be “Serum free homocysteine” and of figure 1G “Liver free homocysteine”.
  3. In line 220 the sentence beginning “Significant lower levels of homocysteine…” should be completed.

Author Response

The authors would like to thank the reviewer for his positive comments and constructive feedback. We agree with all comments and attempted to address them fully. Specifically,

  1. The number of animals per group is small (n = 5 or 4), which might lead to an overrepresentation of high or low metabolite values of individuals. Therefore, please present the data in figure 1B – 1I and in figure 3B – 3K as box plots, which allow the illustration of all individual values in each group.

Response: We agree entirely with the reviewer. We modified the display of the figures to show individual values.

  1. The heading of figure 1C should be “Serum free homocysteine” and of figure 1G “Liver free homocysteine”.

Response: We agree and made the modifications suggested

  1. In line 220 the sentence beginning “Significant lower levels of homocysteine…” should be completed.

Response: The reviewer is correct that a verb was missing in this sentence. Please find the new phrasing highlighted in yellow on line #229

Reviewer 2 Report

This manuscript studied the impact of methionine restriction on its metabolites concentration in serum and liver, hepatic DNA and protein methylation, and gut microbiome of mice, which provided information to health outcomes with dietary nutrients modulation. The manuscript is well prepared and easy to follow for the readers, minor revision is suggested with the following points:

  • Please provide the dietary nutrients of the experimental diets, as the methionine metabolism could be impacted by other nutrients in the diets, for example, cysteine, protein, energy etc.
  • Please illustrate the nutritional status of mice before the trial started. Did they consume the same diet/nutrients?
  • The authors used a two-way ANOVA analysis, therefore, if the interaction is significant, the interaction should be presented, if the interaction is not significant, the main effect should be presented. Also, when the interaction is significant, the P-value of interaction, rather than the pairwise comparison should be provided.
  • In the discussion, only mucinophila and Bacteroidales S24-7 group uncultured bacterium had been discussed, please add the discussion of the other bacteria presented in the results.
  • Please add the discussion between the serum/hepatic methylation status and gut microbiome, and the conclusion of this study.

Author Response

The authors would like to thank the reviewer for her/his excellent suggestions. We agree with all comments and attempted to address them fully. Specifically,

  1. Please provide the dietary nutrients of the experimental diets, as the methionine metabolism could be impacted by other nutrients in the diets, for example, cysteine, protein, energy etc.

Response: We agree that this information is useful to the reader. We included the catalog numbers (Lines 77-78) as well as a supplementary table listing the precise composition of the control and methionine restricted diet (Supplementary Table S1)

  1. Please illustrate the nutritional status of mice before the trial started. Did they consume the same diet/nutrients?

Response: We apologize for omitting this information in the original draft. Yes, mice were all fed an identical diet up to the start of the experiment. This information is now provided in lines 73-74

  1. The authors used a two-way ANOVA analysis, therefore, if the interaction is significant, the interaction should be presented, if the interaction is not significant, the main effect should be presented. Also, when the interaction is significant, the P-value of interaction, rather than the pairwise comparison should be provided.

Response: We would like to thank the reviewer for this suggestion. We agree that this information is very useful to the reader. We added these parameters to figures 1 and 3 and made slight modifications to section 3.3 to reflect this information. The changes are highlighted in yellow between lines 193 and 205.

  1. In the discussion, only mucinophila and Bacteroidales S24-7 group uncultured bacterium had been discussed, please add the discussion of the other bacteria presented in the results.

Response: We would like to thank the reviewer for this suggestion. We believe that the readers will appreciate the comments we added on the association between the ratio of Bacteroidetes (such as Bacteriodaceae) and Firmicutes (such as Ruminococcaceae) and obesity in humans (lines 256-261).

  1. Please add the discussion between the serum/hepatic methylation status and gut microbiome, and the conclusion of this study.

Response: We would like to thank the reviewer for this suggestion. There are indeed interesting correlations between the composition of the gut microbiome and gene-specific DNA methylation. We hope that the reviewer will appreciate the comments to this effect that we included in lines 262-266.